# Brain Injury Is Prevalent and Precedes Tobacco Use among Youth and Young Adults Experiencing Homelessness

**DOI:** 10.3390/ijerph20065169

**Published:** 2023-03-15

**Authors:** Julianna M. Nemeth, Allison M. Glasser, Alice Hinton, Joseph M. Macisco, Amy Wermert, Raya Smith, Hannah Kemble, Georgia Sasser

**Affiliations:** 1Division of Health Behavior and Health Promotion, College of Public Health, The Ohio State University, Columbus, OH 43210, USA; 2Division of Biostatistics, College of Public Health, The Ohio State University, Columbus, OH 43210, USA; 3College of Arts and Sciences, The Ohio State University, Columbus, OH 43210, USA; 4Division of Health Services, Management, and Policy, College of Public Health, The Ohio State University, Columbus, OH 43210, USA

**Keywords:** acquired brain injury (ABI), traumatic brain injury (TBI), anoxic and hypoxic injury, strangulation, tobacco initiation, homelessness, youth and young adults, domestic violence, intimate partner violence, peer violence

## Abstract

70%+ of youth and young adults experiencing homelessness (YYEH; 14–24 years old) smoke combustible tobacco. Little is known about the prevalence of acquired brain injury (ABI) among youth and young adult smokers experiencing homelessness (YYSEH) and its impact on tobacco use progression—the aim of our study. Through an interviewer-administered survey, YYSEH were asked about timing of tobacco use; exposure to causes of ABI; including brain oxygen deprivation (BOD; strangulation; accidental; choking games) and blunt force head trauma (BFHT; intentional; shaken violently; accidental); and perpetrators of intentional assault. Participants (*n* = 96) were on average 22 years old and from populations who experience structural disparities; including those minoritized by race (84.4%) and gender/sexual orientation (26.0%). In total, 87% of participants reported at least one exposure to BFHT and 65% to BOD. Intentional injury was more common than accidental. Furthermore, 60.4% of participants (*n* = 59) were classified as having ABI using the Brain Injury Severity Assessment. A significant proportion of YYSEH living with ABI were exposed to both BFHT and BOD prior to trying (68.5%, *p* = 0.002) and to first regular use (82.8%, *p* < 0.001) of tobacco. Among YYSEH with ABI; injury exposure occurred a median of 1 and 5 years before age of first regular tobacco use, dependent on injury mechanism. ABI from intentional violence is prevalent and precedes tobacco use among YYSEH.

## 1. Background

Acquired brain injury (ABI) is a known risk in adults experiencing homelessness, with an estimated prevalence of 88% [1]. While this phenomenon has been studied more extensively in adults, the prevalence of brain injury in youth and young adults experiencing homelessness (YYEH; aged 14–24) has not been evaluated. Brain injury is, in turn, known to be associated with the use of tobacco and other substances [2,3]. Prior literature has established an increased prevalence of tobacco use among YYEH, but to date there has been no analysis on its comorbidity with brain injury in this population. Therefore, we intended to establish the prevalence of ABI and associated tobacco use among YYEH.

ABI includes traumatic brain injury (TBI) from blunt force head trauma events, along with hypoxic events, stroke or opioid overdose, toxic events, and infectious processes [1] that result in brain oxygen deprivation, and is defined by the Commission on Accreditation of Rehabilitation Facilities as “an insult to the brain that affects its structure or function, resulting in impairments of cognition, communication, physical function, or psychosocial behavior” and “does not include brain injuries that are congenital, degenerative, or induced by birth trauma” [4]. Most data capture TBI (vs. events resulting in oxygen deprivation), which disproportionately affects young people and is a leading cause of death among children and adolescents in high-income countries [5,6,7]. An estimated 475,000 children aged 0–14 years sustain TBIs annually in the United States (US) [7]. YYEH are a highly vulnerable subset of youth experiencing health disparities in relationship to stably housed adolescents and young adults, so it would be anticipated that levels of ABI among YYEH would be higher than among youth in general. Despite this, there is little data on the prevalence and impact of ABI among YYEH, with most studies focusing on ABI among adults experiencing homelessness. Only one study reported the prevalence of TBI in adolescents and young adults experiencing homelessness, which was 43%, and of those, 51% experienced their first TBI prior to or at the same age as (10%) their first homeless episode [5]. Although assessing TBI among adults experiencing homelessness, a systematic review and meta-analysis found that the average age at which the first TBI is experienced in the homeless population is 15.8 years [8].

In a 2020 systematic review of the evidence on TBI in homeless and marginally housed individuals, lifetime TBI prevalence amongst homeless and marginally housed individuals is estimated at 53%, five times higher than estimates for the general population [7]. While there is ample analysis of data on TBIs in the homeless population, there are few data on the prevalence of other types of brain injury, such as anoxic and hypoxic injuries. In a report prepared by Conroy and colleagues in 2013, the most common type of ABI among older adults experiencing homelessness was alcohol-related brain injury (86%), followed by TBI, hypoxic events, and infectious processes, which is likely related to the high prevalence of cognitive impairment later in life [1]. ABIs are associated with psychosocial problems, such as relationship breakdown, substance abuse, homelessness, social exclusion, poverty, marginalization, and seizures [5,6,7]. TBI in the population experiencing homelessness is associated with physical health problems, mental health problems, suicidality, and mortality [8]. ABI can be both a cause and consequence of homelessness. The physical, mental, and psychosocial effects of brain injury provide challenges in obtaining and remaining in stable housing [7], and pre-existing mental health conditions and substance use increase the risk of ABI and therefore homelessness [7]. Furthermore, there is a dose-dependent relationship between the duration of homelessness and occurrence of ABI, meaning the longer someone remains homeless, the higher their risk of ABI [7]. 

There may also be a reciprocal relationship between ABI and substance use. In an interpretative phenomenological review on ABI, substance use, and homelessness, data from eleven studies from the US reported two-thirds of individuals with ABI in rehabilitation had histories of substance abuse prior to the onset of ABI [6]. Furthermore, a history of ABI is bidirectionally associated with substance abuse problems, in that substance use increases the risk of ABI, while sometimes substance abuse only happens once individuals become homeless following an ABI [2]. Several studies have found an increased risk of smoking among those who experienced an ABI. In a 2011 population-based cross-sectional study of Canadian adolescents, high schoolers who reported having a TBI in their lifetime had odds 2.5 times greater for daily smoking relative to those without TBI [3]. The same data set showed that sex moderated the relationship between lifetime TBI and cigarette smoking. Late adolescent males with TBI had greater odds of daily smoking than their female counterparts [9]. These findings were corroborated by a 2017 longitudinal cohort study of participants less than 17 years of age that found those with TBI had increased odds of problematic use of tobacco and cannabis relative to those without injury [10]. Furthermore, a 2020 analysis of a large cross-sectional survey conducted in 2014 found that after adjusting for demographics, lifetime history of TBI was associated with smoking and that Ohioans who have sustained at least one TBI with loss of consciousness in their lifetime are at increased risk for smoking [11]. However, in a cross-sectional survey of Canadians ages 12+, there was no significant association between TBI and being a current smoker [12].

YYEH also experience adverse childhood experiences (ACEs) at a disproportionate rate of 85.5% (when unaccompanied by family) compared to 34.1% of housed youth [13]. This gap leaves YYEH more vulnerable to associated issues such as substance use. When accounting for both direct and environmental ACEs, a Nebraska-wide survey indicated 45.5% of residents with ACE history smoked cigarettes [14]. ACE poly-victimization has also been found to be positively linked to increased substance use, with Shin (2018) establishing that young adults with history of multiple ACEs reported more substance use, including current tobacco use, than those with fewer/less severe ACEs [15]. Literature also reveals current, severe tobacco use disorders to be associated with having a history of ACEs among participants who experienced youth homelessness [16].

Upwards of 70% of YYEH smoke combustible tobacco [17], and age of initiation is as young as ten years [18]. Given the burden of smoking in this population, it is critical to prevent smoking initiation and to provide support for smoking cessation. Smoking initiation among the general population of youth is associated with the use of other substances [19,20,21] and exposure to smoking in one’s social environment, including among family members [22,23,24]. In addition, smoking initiation is associated with depressive symptoms, low risk perceptions and beliefs that smoking can calm one down when angry or nervous [22,23,24]. Although the evidence suggests that ABI, as well as ACEs, are associated with substance use, and the prevalence of ABI and smoking is much higher among people experiencing homelessness, it is unknown whether early exposure to ABI is associated with smoking initiation. In the process of developing a targeted combustible tobacco cessation intervention for youth and young adults experiencing homelessness, we attempted to characterize the extent of ABI in the population, anticipating that unmet need regarding the treatment of brain injury could be a barrier for accessing supported tobacco-cessation intervention. The purpose of this paper is: (1) to report our findings regarding brain injury prevalence in a purposeful sample of YYEH who smoke tobacco products, as well as to describe the source of the intentionally inflicted injury, and (2) to assess the relationship between age at initiation of tobacco and age at first exposure to events that can lead to ABI (e.g., traumatic head injury and oxygen deprivation events, such as strangulation and overdose).

## 2. Methods

### 2.1. Participants

Study participants were youth and young adults (14 to 24 years of age) experiencing homelessness attending a drop-in center in a Midwestern city. Of the 139 participants recruited for this study, 96 met eligibility criteria and agreed to participate. Eligibility criteria included having used at least one combustible tobacco product in the past week, not currently making an attempt to quit smoking, attending a drop-in center, and not participating in an earlier phase of this study. We also separately analyzed a subset of participants who had experienced a brain injury (*n* = 59).

### 2.2. Measures

#### 2.2.1. Demographic Characteristics

We measured participants’ age, sex assigned at birth (female, male, intersex), gender identity (female, male, genderqueer, transgender female, transgender male, transgender, other), sexual orientation (heterosexual/straight, gay, lesbian, bisexual, queer/questioning, asexual, other), race (American Indian or Alaska Native, Asian, Black or African American, Native American, Native Hawaiian or another Pacific Islander, White, bi- or multi-racial, other), Hispanic ethnicity, and education (less than high school, high school diploma, GED, more than high school). We also assessed adverse childhood problems that could be related to current life experiences (abuse, household challenges, and neglect) using the Behavioral Risk Factor Surveillance System (BRFSS) Adverse Childhood Experiences (ACE) for participants 18 years of age or older [25].

#### 2.2.2. Blunt Force Head Trauma Exposure

Exposure to blunt force trauma to the head was measured in three ways: (1) “How many times in your life have you ever been hit in the head or were made to have your head hit another object by another person?” (Hit in Head), (2) “How many times in your life have you ever been shaken violently?” (Shaken Violently), and (3) “How many times in your life has your head accidently been hurt, like through a traffic accident, sports, or an accidental fall?” (Accidentally Hurt). Response options were never, once, a few times, and too many times to remember. A blunt force head trauma exposure was defined as responding “once”, “a few times”, or “too many times to remember” to at least one of the three items.

#### 2.2.3. Brain Oxygen Deprivation Exposure

We assessed exposure to three different sources of brain oxygen deprivation. First, we assessed exposure to brain oxygen deprivation through choking games by asking, “We understand that young people sometimes play choking games or “pass out” challenges, either by themselves or with one another, to feel high. How many times in your life have you been purposefully “choked out”, with the goal of feeling high?” (Choking Games). Second, we assessed exposure to brain oxygen deprivation through strangulation by asking, “We also know that sometimes someone else might intentionally harm another person by choking them or making them feel like they couldn’t breathe in order to scare, harm or control them. How many times in your life have you ever been choked, strangled or made to feel like you couldn’t breathe by another person trying to scare, harm, or control you?” (Intentional Choking). Third, we assessed exposure to brain oxygen deprivation through accidental events by asking, “How many times in your life have you stopped breathing on accident—this could be from an accident, drug overdose, or medical condition?” (Stopped Breathing on Accident). Response options were never, once, a few times, and too many times to remember. An exposure to brain oxygen deprivation was defined as responding “once”, “a few times”, or “too many times to remember” to at least one of the three items.

#### 2.2.4. Altered Consciousness/Brain Injury

Among those who reported exposure to brain oxygen deprivation, we assessed whether they had experienced altered consciousness, indicating a brain injury, using the Brain Injury Severity Assessment (BISA) [26]. Participants were asked, “Whether intentionally or by accident, when people get hurt in the head or neck, or when people have trouble breathing or stop breathing, they may have symptoms afterwards. After anytime your head was hurt or you had difficulty breathing…” (1) “Did you ever black out or lose consciousness?”, (2) “Did you ever feel dazed or confused or disoriented?”, (3) “Did you ever have memory loss about what happened?”, (4) “Did you ever see stars or spots?”, and (5) “Did you ever feel dizzy?” Response options were never, once, a few times, and too many times to remember. We classified participants as having a brain injury if they reported brain oxygen deprivation and an altered consciousness event. We did not assess for BISA for participants who reported blunt force head trauma exposure; however, among those with a brain injury, all but one participant reported blunt force head trauma exposure in addition to brain oxygen deprivation exposure. 

#### 2.2.5. Age at First Exposure to Events That Can Cause Brain Injury

Participants who reported at least one exposure to brain oxygen deprivation or blunt force head trauma, either intentional or accidental, were asked their age at the first injury. These items included, “How old were you the first time you were either made to feel like you couldn’t breathe or when you stopped breathing on accident?” and “At what age was your head first hit or hurt or were you shaken violently?”.

#### 2.2.6. Age at First Tobacco Use

Participants were asked at what age they first tried tobacco: “How old were you when you tried your first tobacco product?” They were also asked at what age they first used tobacco regularly: “How old were you when you first started using tobacco regularly?”.

### 2.3. Statistical Analyses

Categorical variables were summarized with frequencies and percentages while continuous variables were summarized with means and standard deviations (SD) or medians and interquartile ranges (IQR), as appropriate. Among the full sample, associations between exposure to brain oxygen deprivation and blunt force head trauma were evaluated with Fisher exact tests. The relationship between exposure to blunt force head trauma/brain oxygen deprivation and the age of tobacco initiation was assessed through both univariable and multivariable linear regression where all multivariable models controlled for race and ACE. Among participants who experienced a brain injury, chi-square goodness of fit tests were then used to determine if the proportion of participants who tried tobacco prior to their brain injury was significantly different from those who experienced a brain injury before trying tobacco. The relationship between exposure to blunt force head trauma/brain oxygen deprivation and age of first regular tobacco use was analyzed similarly. All analyses were conducted in SAS 9.4 (SAS Institute Inc., Cary, NC, USA) and *p*-values less than 0.05 were considered to be statistically significant.

## 3. Results

### 3.1. Participant Characteristics

Participants in this study were an average of 22 years old (Table 1) and are representative of YYEH, including a high proportion of those who experience structural inequities compared to the general population of comparable youth and young adults, such as racial and sexual/gender minoritized youth. Participants reported an average of about four adverse childhood experiences. 

About two-thirds of participants reported exposure to brain oxygen deprivation (Table 2). Among those who experienced a brain injury (*n* = 59), nearly 80% of participants reported exposure to brain oxygen deprivation from intentional injury (choking games or intentional choking). Among those with intentional choking exposure, about 60% reported that they had experienced this from a romantic partner. Eighty-seven percent of participants were exposed to blunt force head trauma. Among those who experienced a brain injury, about 90% of reported intentional blunt force head trauma (hit in the head or shaken violently). Romantic partners and peers were most commonly identified as perpetrators of an intentional blunt force head trauma.

Among participants with brain injury (Table 3), 86.4% of participants reported feeling dazed or confused or disoriented at least once after being hurt in the head or neck, followed by feeling dizzy (84.7%), seeing stars or spots (67.8%), blacking out or losing consciousness (59.3%), and having memory loss about what happened (47.5%).

### 3.2. Co-Occurrence of Exposures That Can Lead to Brain Injury

Participants who have been exposed to blunt force head trauma were significantly more likely to have been exposed to brain oxygen deprivation than those who have not been exposed to blunt force head trauma, *p* < 0.001 (Appendix A). Seventy-three percent of participants exposed to blunt force head trauma have also been exposed to brain oxygen deprivation while only eight percent of participants not exposed to blunt force head trauma were exposed to brain oxygen deprivation. There was a significant association between exposure to blunt force head trauma and intentional choking, *p* = 0.009; however, there were no significant associations with exposure to choking games or accidental choking, *p* = 1.000 and *p* = 0.161, respectively. Of the 12 participants who were not exposed to blunt force head trauma, none had been exposed to choking games or were intentionally choked. Only 1 of the 12 stopped breathing on accident, and this only happened a single time.

Almost all (98%) participants who were exposed to brain oxygen deprivation were also exposed to blunt force head trauma, while only 67% of participants who had not been exposed to brain oxygen deprivation were exposed to blunt force head trauma (Appendix A). Specifically, being exposed to brain oxygen deprivation was significantly associated with being hit in the head and shaken violently (*p* < 0.001 for both). However, there was no significant association with being exposed to accidental head trauma.

### 3.3. Age at Tobacco Initiation and Exposures That Can Lead to Brain Injury

The mean age when participants first tried tobacco was 14.2 years (SD = 3.7), and the mean age when participants first regularly used tobacco was 16.8 (SD = 2.5) (Table 1). On average, participants were 10.7 years (SD = 5.8) of age when they were first exposed to blunt force head trauma and 14.1 years (SD = 5.6) of age when first exposed to brain oxygen deprivation. 

Among the full sample of participants, exposures that can lead to brain injury and the age when they first tried tobacco were not significantly associated (Appendix A). However, participants with a brain oxygen deprivation exposure began using tobacco regularly approximately one and a half years earlier than those without a brain oxygen deprivation exposure (median: 1.42 years; 95% CI: −2.42, −0.41; *p* = 0.006) (Figure 1; Appendix A). After adjusting for race and adverse childhood experiences, this association was no longer significant.

Among participants with blunt force head trauma, the age of tobacco initiation occurred a median of 4 years after age of first exposure to blunt force head trauma, while it occurred slightly before the age at first exposure to brain oxygen deprivation (median 1 year) (Figure 2). Median age of first regular use of tobacco was 5 and 1 years after exposure to blunt force head trauma and brain oxygen deprivation, respectively.

A significantly larger proportion of participants who have experienced a brain injury tried tobacco before exposure to brain oxygen deprivation (*p* = 0.014; Table 4). However, there were significantly larger proportions who were exposed to blunt force head trauma or any form of exposure before trying tobacco for the first time. Similarly, there were significantly larger proportions of participants who were exposed to blunt force head trauma or any form of exposure before regularly using tobacco.

## 4. Discussion

In this study, we examined exposures that can lead to brain injury among YYEH who use combustible tobacco. These exposures were prevalent in our sample, with 64.9% being exposed to lifetime oxygen deprivation to the brain and 87.4% exposed to lifetime blunt force trauma to the head—with intentional injury being more common than accidental. Over 60% of our sample were classified as having probable brain injuries as they endorsed at least one type of alteration in consciousness following such events. This is higher than another study of YYEH that found that 43% of their sample had experienced lifetime TBI, but the study only assessed blunt force trauma and did not ask about brain oxygen deprivation [5]. Both studies suggest a much higher prevalence of brain injury among YYEH compared to the general population in the US, with prevalence of lifetime brain injury ranging from 6.5% to 18.3% among youth ages 13–17 years, with variation in estimates across surveys likely attributed to question wording [27].

The majority of those in our study who were exposed to possible brain injury reported co-occurrence of exposure to blunt force head trauma and brain oxygen deprivation. Among those with a co-occurrence of these two types of exposures, one was significantly associated with intentional injury of the other. There is very little published evidence to which we can compare these findings. Violence is a common occurrence for individuals experiencing homelessness. Adults experiencing homelessness are more likely to be treated in trauma centers with intentional violent injuries compared with housed adults [28], and almost three-quarters of YYEH report lifetime intimate partner violence [29]. Intimate partner violence is a predictor of housing instability [30], so experiencing violent injury may have even been a reason for becoming homeless for these youth. Additional research is needed to understand the extent of brain injury, including the types of injury involved, and the role of brain injury in experiences related to homelessness for young people.

In this study we also found that the youth were more likely to be exposed to possible brain injury prior to the age at which they first began regularly using tobacco, although this association was weaker for exposure to brain oxygen deprivation. These results are unsurprising given that the average age at first tobacco use in both our sample and globally representative studies is in the later teen years [31]. Given that the use of tobacco in adolescents is often associated with subsequent other substance use [32], we theorize that comorbid substance use and brain injury could be related to lifetime cannabis, opioid, and other substance use risks in this population. Future research and interventions aimed at YYEH should account for these associated risks and examine ways to address them concurrently. 

In addition, a recent systematic review concluded that nicotine may aid in the recovery of cognitive deficits from brain injury [33]. This potential therapeutic effect, coupled with smoking to cope with trauma and violence and to regulate emotions [34], may contribute to the elevated use of tobacco among YYEH. Therefore, early identification and treatment of brain injury may actually prevent tobacco use initiation among YYEH. In addition, the use of nicotine replacement therapy along with developing methods to accommodate symptoms of brain injury (e.g., poor self-regulation, executive function dysregulation) may also be necessary to improve tobacco users’ ability to access, adhere to, and find success with cessation programs and reduce tobacco use disparities among YYEH.

This study has several limitations to report. First, it is cross-sectional, so the temporality of brain injury and psychosocial factors cannot be established. However, we did ask for age of first exposure to events which lead to brain injury along with age at first use and first regular use of tobacco. Our primary analysis was based on this temporal data. Second, as the COVID-19 pandemic required, we closed study recruitment in March of 2020, resulting in a smaller than originally anticipated convenience sample of YYEH who use combustible tobacco from one drop-in center in one geographic area. Given the sample size limitations there is the possibility of type 1 error, and our findings should not be generalized to other populations of YYEH. Third, data were collected through participant self-report, so measurement error is possible—especially as validated self-reported measures of brain injury from violence in community settings do not yet exist. However, it should be noted that the lead researcher on our study team does have experience collecting violence exposure and behavior data from adolescents using recall methods similar to those used here [35,36]. Fourth, as we did not collect data regarding duration or episodic periods of homelessness, we were unable to examine homelessness in relation to smoking behavior or exposure to blunt force head trauma or oxygen deprivation events and subsequent brain injury. We recognize that the experience of homelessness can cofound both injury exposure and tobacco use data in analysis, so collecting the nature of unstable housing over the course of the adolescent years would be helpful in future research and targeting of services with the population. Given the study limitations, we suggest these findings be interpreted with caution and this work should be validated in larger, more representative samples in future studies. 

Despite the limitations, this is the first study to assess hypoxic and anoxic injury caused by both accidental and intentional oxygen deprivation events (e.g., strangulation) among YYEH and the source of intentional injury, both traumatic, hypoxic/anoxic. Our findings contribute to illuminating a crisis that has remained invisible in public health discourse, brain injury research, and clinical practice in shelters and drop-in centers—partner inflicted brain injury, and brain injury caused by family and other community members [37]. In addition, ours is one of the first studies to illuminate the possible role of brain injury from violence as a mechanism impacting modifiable risk behavior in what is arguably one of the US’s most vulnerable populations—youth and young adults experiencing homelessness—over-representative of populations minoritized by race, gender and sexual orientation. Given the pervasive use of interpersonal violence to assert and maintain structural domination targeted at these populations experiencing the burden of health injustices, our study provides a possible mechanism (e.g., brain injury) that may be contributing both to the elevated prevalence of health risk behaviors and to struggles with safety and health service access, as well as benefit from evidence-based behavioral interventions, including smoking cessation. 

## 5. Conclusions

Our study demonstrated a high prevalence of brain injuries among YYEH as well as the comorbidity of tobacco use with exposure to possible brain injury. Our findings suggest that public health and medical providers who address smoking and other substance use should be aware of the potential influence that brain injury could have on substance use initiation and addiction treatment. Efforts to promote smoking cessation in this population may require the accommodation of brain injury symptoms, such as executive function dysregulation, that could be facilitating smoking and preventing cessation. Our findings also indicate a distinct need for further research into brain injury prevention, screening, subsequent impacts, and services to address brain injuries in this population. 

## Figures and Tables

**Figure 1 ijerph-20-05169-f001:**
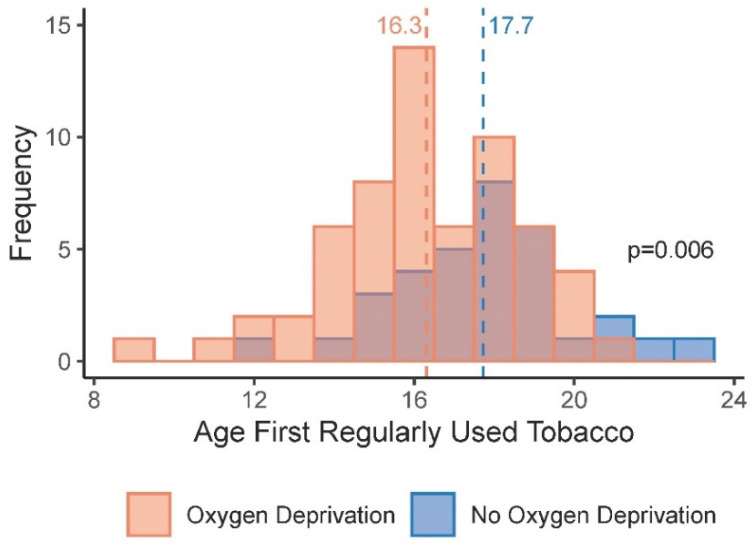
Age at first regular use of tobacco and exposure to brain oxygen deprivation (*n* = 96).

**Figure 2 ijerph-20-05169-f002:**
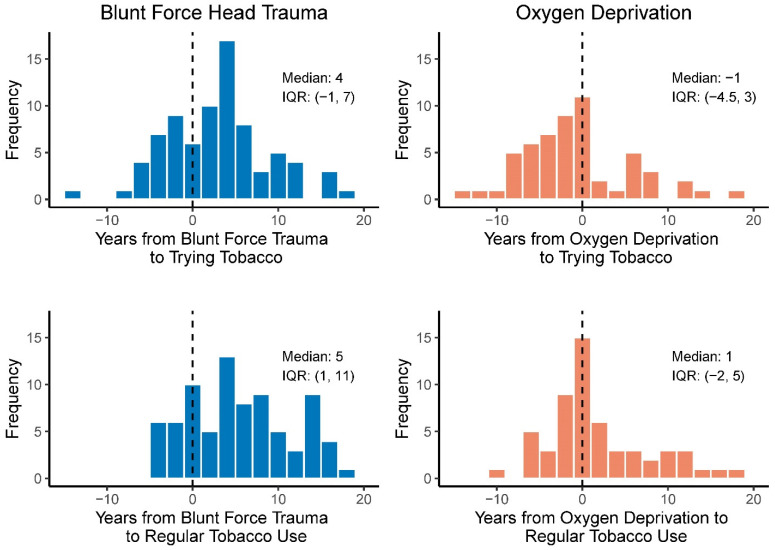
Distribution of time in years from head injury (blunt force trauma exposure or oxygen deprivation exposure) to first trying tobacco and first regular use of tobacco.

**Table 1 ijerph-20-05169-t001:** Participant demographic characteristics.

	All Participants (*n* = 96)		Participants with Brain Injury (*n* = 59)	
Age (mean, SD)	21.82	2.00	21.84	2.12
Gender (*n*, %)				
Cisgender Male	52	54.17	30	50.85
Cisgender Female	39	40.63	25	42.37
Transgender Female	2	2.08	2	3.39
Transgender Male	2	2.08	1	1.69
Non-binary	1	1.04	1	1.69
Race (*n*, %)				
White	15	15.63	12	20.34
Black	51	53.13	27	45.76
Other	30	31.25	20	33.90
Ethnicity (*n*, %)				
Non-Hispanic	88	91.67	54	91.53
Hispanic	8	8.33	5	8.47
Sexual Orientation (*n*, %)				
Heterosexual/Straight	71	73.96	42	71.19
Bisexual	19	19.79	13	22.03
Other	6	6.25	4	6.78
Education (*n*, %)				
Less than High School	31	32.29	22	37.29
High School Diploma	46	47.92	28	47.46
GED	4	4.17	1	1.69
More than High School	15	15.63	8	13.56
Where Slept Most Nights (*n*, %)				
With family or friends/Own home	31	32.29	19	32.20
Shelter/Drop-in-center	27	28.13	18	30.51
Group home/Treatment facility/Detention facility	13	13.54	7	11.86
Outside/Car/Tent	25	26.04	15	25.42
BRFSS ACE (mean, SD)	4.23	2.65	4.84	2.59
Age First Tried Tobacco (mean, SD)	14.15	3.69	13.81	3.57
Age First Regularly Used Tobacco (mean, SD)	16.77	2.46	16.39	2.38

BRFSS: Behavioral Risk Factor Surveillance System; ACE: adverse childhood experience.

**Table 2 ijerph-20-05169-t002:** Brain oxygen deprivation and blunt force head trauma exposure.

	All Participants (*n* = 96)		Participants with Brain Injury (*n* = 59)	
**Brain Oxygen Deprivation** **Exposure (*n*, %)**				
None	33	35.11	0	0.00
Any	61	64.89	59	100.00
Intentional Oxygen Deprivation Exposure (Choking Games or Intentional Choking) (*n*, %)				
None	46	48.42	12	20.34
Any	49	51.58	47	79.66
Choking Games (*n*, %)				
Never	84	87.50	48	81.36
Once	5	5.21	5	8.47
A few times	4	4.17	4	6.78
Too many times to remember	2	2.08	2	3.39
Refuse	1	1.04	0	0.00
Intentional Choking (*n*, %)				
Never	51	53.13	17	28.81
Once	12	12.50	11	18.64
A few times	25	26.04	24	40.68
Too many times to remember	7	7.29	7	11.86
Refuse	1	1.04	0	0.00
Stopped Breathing on Accident (*n*, %)				
Never	53	55.21	18	30.51
Once	18	18.75	18	30.51
A few times	17	17.71	17	28.81
Too many times to remember	4	4.17	4	6.78
Don’t Know	3	3.13	2	3.39
Refuse	1	1.04	0	0.00
People Responsible for Intentional Oxygen Deprivation Exposure (*n*, %) ^a^				
Parent/Guardian	11	26.19	11	27.50
Brother or Sister	12	28.57	11	27.50
Other Family Member	8	19.05	8	20.00
Romantic Partner	25	59.52	25	62.50
Peer	12	29.27	12	30.00
Other	7	16.67	6	15.38
Age of First Brain Oxygen Deprivation Exposure (mean, SD)	14.11	5.63	13.90	5.59
Most Recent Brain Oxygen Deprivation Exposure (*n*, %)				
Past 3 Days	3	4.92	3	5.08
Past Month	7	11.48	7	11.86
Past Year	15	24.59	15	25.42
Longer than a Year ago	33	54.10	31	52.54
Don’t Know	3	4.92	3	5.08
Brain Oxygen Deprivation Exposure Prior to First Trying Tobacco (*n*, %)				
No	38	67.86	36	66.67
Yes	18	32.14	18	33.33
Brain Oxygen Deprivation Exposure Prior to First Regularly Using Tobacco (*n*, %)				
No	25	44.64	23	42.59
Yes	31	55.36	31	57.41
**Blunt Force Head Trauma** **Exposure (*n*, %)**				
None	12	12.63	1	1.69
Any	83	87.37	58	98.31
Intentional Blunt Force Head Trauma Exposure (Hit in Head of Shaken Violently) (*n*, %)				
None	25	26.60	6	10.34
Any	69	73.40	52	89.66
Hit in Head (*n*, %)				
Never	29	30.21	9	15.25
Once	10	10.42	7	11.86
A few times	36	37.50	27	45.76
Too many times to remember	20	20.83	16	27.12
Refuse	1	1.04	0	0.00
Shaken Violently (*n*, %)				
Never	56	58.33	24	40.68
Once	6	6.25	6	10.17
A few times	19	19.79	15	25.42
Too many times to remember	13	13.54	13	22.03
Don’t Know	1	1.04	1	1.69
Refuse	1	1.04	0	0.00
Accidentally Hurt (*n*, %)				
Never	29	30.21	12	20.34
Once	20	20.83	14	23.73
A few times	32	33.33	23	38.98
Too many times to remember	14	14.58	10	16.95
Refuse	1	1.04	0	0.00
People Responsible for Intentional Blunt Force Head Trauma Exposure (*n*, %) ^a^				
Parent/Guardian	22	33.33	20	40.00
Brother or Sister	22	32.84	20	40.00
Other Family Member	12	17.91	9	18.00
Romantic Partner	34	50.75	28	56.00
Peer	39	59.09	27	54.00
Other	14	20.90	9	18.37
Age of First Blunt Force Head Trauma Exposure (mean, SD)	10.65	5.76	10.78	5.51
Most Recent Blunt Force Head Trauma Exposure (*n*, %)				
Past 3 Days	4	4.82	2	3.45
Past Month	11	13.25	9	15.52
Past Year	28	33.73	22	37.93
Longer than a Year ago	39	46.99	24	41.38
Don’t Know	1	1.20	1	1.72
Blunt Force Head Trauma Exposure Prior to First Trying Tobacco (*n*, %)				
No	24	30.38	17	31.48
Yes	55	69.62	37	68.52
Blunt Force Head Trauma Exposure Prior to First Regularly Using Tobacco (*n*, %)				
No	17	21.52	12	22.22
Yes	62	78.48	42	77.78
Type of Intentional Exposure (*n*, %)				
None	23	24.47	4	6.90
Brain Oxygen Deprivation Exposure Only	2	2.13	2	3.45
Blunt Force Trauma Exposure Only	23	24.47	8	13.79
Both Exposures	46	48.94	44	75.86
Exposure That Can Lead to Brain Injury Prior to First Trying Tobacco (*n*, %)				
Both Exposures	16	18.82	14	24.14
Brain Oxygen Deprivation Exposure Only	4	4.71	4	6.90
Blunt Force Trauma Exposure Only	41	48.24	23	39.66
Neither	24	28.24	17	29.31
Exposure That Can Lead to Brain Injury Prior to First Regularly Using Tobacco (*n*, %)				
Both Exposures	25	30.12	25	43.10
Brain Oxygen Deprivation Exposure Only	6	7.23	6	10.34
Blunt Force Trauma Exposure Only	37	44.58	17	29.31
Neither	15	18.07	10	17.24

^a^ Only asked of participants at least 18 years of age.

**Table 3 ijerph-20-05169-t003:** Characteristics of brain injury (those who experienced altered consciousness) (*n* = 59).

	Participants with Brain Injury (*n* = 59)	
Brain Injury Severity Assessment (BISA) Item (*n*, %)		
Black out or lose consciousness (*n*, %)		
Never	23	38.98
Once	11	18.64
A few times	20	33.90
Too many times to remember	4	6.78
Don’t Know	1	1.69
Feel dazed or confused or disoriented (*n*, %)		
Never	8	13.56
Once	20	33.90
A few times	25	42.37
Too many times to remember	6	10.17
Have memory loss about what happened (*n*, %)		
Never	31	52.54
Once	9	15.25
A few times	14	23.73
Too many times to remember	5	8.47
See stars or spots (*n*, %)		
Never	19	32.20
Once	11	18.64
A few times	23	38.98
Too many times to remember	6	10.17
Feel dizzy (*n*, %)		
Never	9	15.25
Once	15	25.42
A few times	24	40.68
Too many times to remember	11	18.64

BISA: Brain Injury Severity Assessment.

**Table 4 ijerph-20-05169-t004:** Time from exposure to tobacco initiation or regular use among the subset of participants who experienced altered consciousness (with a brain injury); *p*-values from chi-square tests for equal proportions.

Age Tried Tobacco	*n*	%	*p*-Value
Brain Oxygen Deprivation Exposure (*n* = 54)			0.014
Tried Tobacco Before Exposure	36	66.67	
Exposure Before Tried Tobacco	18	33.33	
Blunt Force Head Trauma Exposure (*n* = 54)			0.007
Tried Tobacco Before Exposure	17	31.48	
Exposure Before Tried Tobacco	37	68.52	
Brain Oxygen Deprivation or Blunt Force Head Trauma Exposure (*n* = 58)			0.002
Tried Tobacco Before Exposure	17	29.31	
Exposure Before Tried Tobacco	41	70.67	
**Age First Regularly Used Tobacco**			
Brain Oxygen Deprivation Exposure (*n* = 54)			0.276
Use Tobacco Before Exposure	23	42.59	
Exposure Before Use Tobacco	31	57.41	
Blunt Force Head Trauma Exposure (*n* = 54)			<0.001
Use Tobacco Before Exposure	12	22.22	
Exposure Before Use Tobacco	42	77.78	
Brain Oxygen Deprivation or Blunt Force Head Trauma Exposure (*n* = 58)			<0.001
Use Tobacco Before Exposure	10	17.24	
Exposure Before Use Tobacco	48	82.76	

## Data Availability

The data presented in this study are available on request from the corresponding author. The data are not publicly available due to ongoing analyses by the study team.

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
