# Peer review of "Brain Injury Is Prevalent and Precedes Tobacco Use among Youth and Young Adults Experiencing Homelessness"

_ijerph, 2023, doi:10.3390/ijerph20065169_

Round 1

Reviewer 1 Report

This is an interesting study on acquired brain injury and the impact on tobacco use initiation and progression. The findings certainly add important findings to the existing body of knowledge on brain injury among populations experiencing homelessness and in particular youth and young adults.

There is one key area that seems to be missing and does not seem to be incorporated into the study, and that is the duration of homelessness for these YYSEH, and how that may confound both injury exposure and tobacco use, and if the authors collected that data, would be great to include. If that information is not available, then it's likely worth expanding this point in the discussion of limitations. Although the authors mention that there was no attempt to determine the relationship between the variables and likelihood of experiencing homelessness, this seems to be different than the duration of homelessness on both risk factors and outcomes (use of tobacco). 

Other than that, the study was well conducted, the associations are strong enough to stand on their own even though there were multiple comparisons made and adjustments were not clearly described, and the paper is well written.

Author Response

Reviewer 1: This is an interesting study on acquired brain injury and the impact on tobacco use initiation and progression. The findings certainly add important findings to the existing body of knowledge on brain injury among populations experiencing homelessness and in particular youth and young adults. 

Overall Response to Reviewer 1:  Thank you for your review of our article, and for sharing your overall support of our study. 

Point 1: There is one key area that seems to be missing and does not seem to be incorporated into the study, and that is the duration of homelessness for these YYSEH, and how that may confound both injury exposure and tobacco use, and if the authors collected that data, would be great to include. If that information is not available, then it's likely worth expanding this point in the discussion of limitations. Although the authors mention that there was no attempt to determine the relationship between the variables and likelihood of experiencing homelessness, this seems to be different than the duration of homelessness on both risk factors and outcomes (use of tobacco).  

Response to Point 1: The study did not collect any information regarding the duration of homelessness; although we appreciate that this would be valuable information to investigate in future studies. We did, however, collect information about where participants sleep most nights. While this does not provide any information about the duration the youth have had unstable housing it does help to provide a more complete picture of their current housing situation. This variable has now been added to Table 1. We also added the following to the limitations section: 

Fourth, as we did not collect data regarding duration or episodic periods of homelessness, we were unable to examine homelessness in relation to smoking behavior or exposure to blunt force head trauma or oxygen deprivation events and subsequent brain injury.  We recognize that the experience of homelessness can cofound both injury exposure and tobacco use data in analysis, so collecting the nature of unstable housing over the course of the adolescent years would be helpful in future research and targeting of services with the population. 

Point 2: Other than that, the study was well conducted, the associations are strong enough to stand on their own even though there were multiple comparisons made and adjustments were not clearly described, and the paper is well written. 

Response to Point 2: Thank you for the positive comments regarding the conduct of our study.  While there were many separate statistical tests conducted, you are correct in your assumption that no adjustments were made for multiple comparisons. This was done as this is a secondary analysis on a small sample with the main purpose of generating pilot data and hypotheses for future larger and more thorough studies investigating the associations between acquired brain injury and tobacco use. We wanted to note that the only analysis where we controlled for co-variants is presented in Supplemental Table 3.  In this case, we’ve provided a note at the bottom of the table concerning adjustments.   

Reviewer 2 Report

Thank you for allowing me to review this manuscript. The authors have chosen a topic that is not necessarily one that is addressed often. My take of this manuscript is that we all as health professionals need to see tobacco uptake as a symptom of larger issues that need to be addressed.

Line 132, page 1, indention needs to be decreased

Conclusions: authors bring up the idea that tobacco initiation may be therapeutic of the brain injury. This was the first time I saw this in the manuscript, was that a question that was asked of sample? Please make clearer that this was not something addressed in your study.

Limitations were adequately addressed.

Author Response

Reviewer 2: Thank you for allowing me to review this manuscript. The authors have chosen a topic that is not necessarily one that is addressed often. My take of this manuscript is that we all as health professionals need to see tobacco uptake as a symptom of larger issues that need to be addressed.

Overall Response to Reviewer 2:  Thank you for your thoughtful review of our manuscript and for providing points for modification.

Point 3: Line 132, page 1, indention needs to be decreased

Response to Point 3: The formatting error on line 132 has been corrected.

Point 4: Conclusions: authors bring up the idea that tobacco initiation may be therapeutic of the brain injury. This was the first time I saw this in the manuscript, was that a question that was asked of sample? Please make clearer that this was not something addressed in your study.

Response to Point 4:  You are correct.  We were not asking this question of our sample. We were rather situating our findings within current literature in hopes of illuminating things to consider for practitioners and scientists as we work towards closing the tobacco use gap among this population.  We have revised the paragraph in the conclusions to try to be more clear about this.  The paragraph now reads:

In addition, a recent systematic review concluded that nicotine may aid in recovery of cognitive deficits from brain injury.[33] This potential therapeutic effect, coupled with smoking to cope with trauma and violence and to regulate emotions,[34] may contribute to the elevated use of tobacco among YYEH.  Therefore, early identification and treatment of brain injury may actually prevent tobacco use initiation among YYEH.  In addition, to improve tobacco users ability to access, adhere to, and successfully quit using combusted tobacco, in particular, use of nicotine replacement therapy along with developing methods to accommodate symptoms of brain injury (e.g., poor self-regulation) in the cessation process may be necessary to close tobacco use disparities among YYEH.

Point 5: Limitations were adequately addressed.

Response to Point 5:  Thank you for noting that our study’s limitations were adequately addressed; however, we wanted to note that we did make revisions to our limitations section given feedback from other reviewers.

Reviewer 3 Report

In the last paragraph of Section 4, the authors mention several limitations, such as the lack of causality in the findings due to the cross-sectional data, the sample bias due to the non-representativeness of the sample, and measurement errors due to self-reported data. The authors argue that the study is still publishable just because it is the first study for the topic. However, those limitations are too critical, to the extent they make all the findings and conclusions not trustable. The sample is also too small (n<100) to do any meaningful statistical analysis.   

Author Response

Reviewer 3:

Point 6: In the last paragraph of Section 4, the authors mention several limitations, such as the lack of causality in the findings due to the cross-sectional data, the sample bias due to the non-representativeness of the sample, and measurement errors due to self-reported data. The authors argue that the study is still publishable just because it is the first study for the topic. However, those limitations are too critical, to the extent they make all the findings and conclusions not trustable. The sample is also too small (n<100) to do any meaningful statistical analysis. 

Response to Point 6: Thank you for raising your concerns over the limitations we have highlighted in our own study.  As we know you are aware, no single study is free of limitations.  We do agree that the sample is not representative of the population as a whole, however, no known studies of youth and young adults experiencing homelessness are. The potential for measurement error and the fact that the data was cross-sectional are unavoidable limitations of this type of research—common in this population that is incredibly difficult and costly to study over time as part of a longitudinal cohort without significant funding. We also recognize that the sample size is small, but smaller studies such as this are needed to call attention to this novel area of research (brain injury from intention violence and its association to the leading causes of preventable diseases and early death) before larger more representative studies will be funded and can take place. As a result of the small sample size, we limited the statistical analyses conducted and took care not to overfit models to the data, focusing mainly on univariate associations and only adjusting for 2 variables (race and ACE) in the multivariable models presented in Supplemental Table 3. Of course, these findings will need to be confirmed in a larger more generalizable population, but this study is the first piece of the puzzle.

We have significantly revised both the explanation of our limitations as well as the unique contributions of this study to current practice and future research directions.  As limited work is taking place in this vulnerable population, both in respect to brain injury screening and accommodation along with tobacco use initiation and cessation, we hope that you may be understanding of the importance of publishing formative research such as this despite the limitations. The limitations and contributions paragraphs now read:

This study has several limitations to report. First, it is cross-sectional, so the temporality of brain injury and psychosocial factors cannot be established. However, we did ask for age of first exposure to events which lead to brain injury along with age first used and first regularly used tobacco.  Our primary analysis was based on this temporal data. Second, as the COVID-19 pandemic required, we closed study recruitment in March of 2020, resulting in a smaller than originally anticipated convenience sample of YYEH who use combustible tobacco from one drop-in center in one geographic area.  Given the sample size limitations there is the possibility of type 1 error, and our findings should not be generalized to other populations of YYEH. Third, data were collected through participant self-report, so measurement error is possible—especially as validated self-reported measures of brain injury from violence in community settings do not yet exist.  However, it should be noted that the lead researcher on our study team does have experience collecting violence exposure and behavior data from adolescents using recall methods similar to those used here. [35, 36] Fourth, as we did not collect data regarding duration or episodic periods of homelessness, we were unable to examine homelessness in relation to smoking behavior or exposure to blunt force head trauma or oxygen deprivation events and subsequent brain injury.  We recognize that the experience of homelessness can cofound both injury exposure and tobacco use data in analysis, so collecting the nature of unstable housing over the course of the adolescent years would be helpful in future research and targeting of services with the population. Given study limitations, we suggest these findings be interpreted with caution and this work should be validated in larger, more representative samples in future studies. 

Despite limitations, this is the first study to assess hypoxic-anoxic injury caused by both accidental and intentional oxygen deprivation events (e.g., strangulation) among YYEH and the source of intentional injury, both traumatic and hypoxic-anoxic.  Our findings contribute to illuminating a crisis that has remained invisible in public health discourse, brain injury research, and clinical practice in shelters and drop-in centers--partner inflicted brain injury, and brain injury caused by family and other community members.[37] In addition, ours is one of the first studies to illuminate the possible role of brain injury from violence as a mechanism impacting modifiable risk behavior in what is arguably one of the US’s most vulnerable populations—youth and young adults experiencing homelessness, over-representative of populations minoritized by race, gender and sexual orientation.  Given the pervasive use of interpersonal violence to assert and maintain structural domination targeted at these populations experiencing the burden of health injustices, our study provides a possible mechanism (e.g., brain injury) that may be contributing both to elevated prevalence of health risk behaviors and to struggles with safety and health service access, as well as benefit from evidence-based behavioral interventions, including smoking cessation.   

Round 2

Reviewer 3 Report

None